# An Intangible-Asset Approach to Strategic Business-IT Alignment

**Miguel Tejada-Malaspina \*** and **Alberto Un Jan**

Faculty of Industrial and Systems Engineering, National University of Engineering, Lima 15333, Peru; eun-jan@uni.edu.pe
\* Correspondence: mtejada@uni.edu.pe; Tel.: +51-1-613-2377

**Abstract:** The correct use of information technology (IT) in business is a longstanding critical issue due to the competitive advantages and performance that IT generates when it is managed strategically and correctly aligned with a business' strategies and processes. A conceptual model is presented to investigate the effects of intangible assets and organizational capabilities on business-IT strategic alignment. Social networks between business and IT executives conform to relational capital that permits the creation of combinative capabilities; these capabilities encourage the transfer, integration, learning, and strategic use of business and IT executives' knowledge, and affect the level of strategic business-IT alignment. This combination of social-network characteristics and organizational capabilities in order to generate strategic business-IT alignment is new.

**Keywords:** intangible assets; organizational capabilities; knowledge management; strategic business-IT alignment

## 1. Introduction

In almost all business sectors, information technology (IT) is considered an important tool for achieving performance in a firm, expressed in operational effectiveness, financial profitability, business agility, competitive advantage, and adaptability to a business environment. The presence and importance of IT are constantly growing as technological advances increasingly automate business processes. The introduction and effective use of IT in enterprises are not simple, and often do not achieve the planned objectives. This situation is worse if IT is not managed, or planned and executed, with its strategic aspects in mind.

Since more than three decades ago, factors that affect business-IT alignment have been studied, these factors were classified as intellectual and social factors. In the last 25 years, social factors have been considered more significantly; these social factors were introduced in the resource-based view of the organizations along with intangible assets and new approaches, such as the knowledge-based view and dynamic capabilities were used to characterize some social factors.

The aim of this study is to analyze the alignment of business and IT strategies, and the factors that determine this alignment in the context of intangible assets, knowledge-based view, and dynamic capabilities.

This research is of theoretical interest because it may improve the understanding of how factors combine to affect the process of strategic business-IT alignment. The research problem of characterizing and explaining relationships between intangible assets and the strategic-alignment process has social implications due to the interests of business and IT managers in working in an integrated and effective manner, and in reducing problems, inefficiencies, and disconnected strengths inside the organization, thereby closing the gap between business and IT.



In this section, we present the aims and theoretical and practical interest of this research. Section 2 presents a literature review on alignment between business and IT strategies, focusing on intangible assets and the social dimension of this alignment, and describing the problem of the lack of social factors as an inhibitor of business-IT alignment. Section 3 presents the conceptual model with its constructs, relationships, and operational variables. Section 4 summarizes a discussion about the achieved results. Section 5 concludes this article and discusses future work.

## 2. Literature Review

### 2.1. Social Dimension of Strategic Business-IT Alignment

The alignment between business and IT strategies has been studied for over 30 years. Horovitz [1] described the alignment as having two dimensions, the social and the intellectual dimension. The intellectual dimension comprises methodologies, techniques, metrics, tools, and other analytical and formal frameworks to obtain strategic alignment.

The social dimension consists of the attitudes and skills of the people involved in both sides of the alignment situation, namely, business and IT people. This dimension is related to the behavior of individuals who participate in the alignment process and the different relationships between them, including their level of involvement, the used methods of communication and decision-making, and the extent to which business and IT executives understand and are committed to each other´s missions, objectives, and plans.

Henderson and Venkatraman [2] presented a model of business and IT alignment that had two sides. First, the external side, which represents business and IT strategies composed of three aspects: scope, distinctiveness or competencies, and governance; and second, the internal side, which is composed of the administrative and IT resources needed to carry out strategies proposed by senior managers, like business and IT infrastructure, processes, and personnel skills.

The model also defined a functional integration between the external and internal sides that operationalizes the strategies in the infrastructure, and a strategic fit between business and IT both at the strategic and operational levels.

Strategic alignment is when an organization applies IT in a strategic and timely manner that is also in harmony with business strategies, goals, and needs [3].

Reich and Benbasat [4] defined the social dimension of alignment as the state in which business and IT executives within an organization understand and are committed to business and IT missions, objectives, and plans. Reich and Benbasat performed exploratory research on the social aspects of alignment with semi-structured interviews, and collected business and IT strategic plans in 10 business units to develop a model in which two antecedents and two current practices were inter-related. The two antecedents were (a) shared domain knowledge between business and IT executives, and (b) success in IT implementation. The two current practices were (a) communication between business and IT executives, and (b) connection between business and IT planning.

Communication between business and IT executives is the contact between individuals during the daily course of events: face to face or written, formal or informal, in groups or in pairs. This concept represents all communications between IT and business executives, except for formal long-term planning sessions. The proposition in a previous study [4] is that the level of communication between IT and business executives is positively related to the level of alignment.

The connection between business and IT planning is the degree to which the business and IT planning processes are inter-related; if both processes occur in conjunction, the level of alignment is high. The IT planning process is a crucial time to forge alignment. The proposition in a previous study [4] is that the level of connection between business and IT planning processes is positively related to the level of alignment.

The shared domain knowledge between business and IT executives is the ability of executives to understand and participate in another´s key processes, and to respect their unique contribution and

challenges. The proposition in a previous study [4] is that the level of shared domain knowledge is positively related to communication between business and IT executives, and the connections between business- and IT-planning processes.

De Haes [5] found that IT governance implementation leads to business-IT alignment in large-size financial-service organizations in Belgium, and that IT governance is composed of three elements, structures, processes, and relational mechanisms. The relational mechanisms are composed of business-IT participation, strategic dialogue, training, shared learning, proper communication [6], specifically two-way communication, and participative and collaborative relationships between business and IT people.

De Haes [5] presented a list of IT government practices with respect to relational mechanisms that influence business-IT alignment. These are (a) job rotation, or IT staff working in business units and business people working in IT; (b) colocation, or business and IT people being physically closer to each other; (c) cross-training, or training business people about IT and IT people about business; (d) knowledge management, or systems to share and distribute knowledge about the IT governance framework, responsibilities, and tasks; (e) business-IT account managers bridging the gap between business and IT [5,7,8]; (f) senior business and IT managers setting examples by acting as partners; (g) informal meetings between business and IT senior managers with no agenda and talking about general activities; and (h) IT leadership, or the ability of CIOs to articulate a vision for IT's role and ensuring that this vision is clearly understood by managers throughout the organization [9].

The communication factor for alignment is a common theme in the literature. Kanter [10] found that the most commonly stated attribute contributing to successful planning and business-IT alignment was constant communication between business and IT managers, and their respective staff. Lind and Zmud [11] mentioned that communication frequency and depth are factors in business-IT alignment. Communications skills, along with technical and business knowledge, are suggested to be essential in achieving business and IT alignment [7]. Sledgionowski and Luftman [12] found that communication between business and IT executives is influenced by factors such as frequency, richness, and protocol rigidity, as well as mutual understanding, intra-/inter-organization learning, and knowledge-sharing.

Sabherwal et al. [13] emphasized the social and cultural dimensions of alignment, such as knowledge-sharing and a common strategic vision between IT and business management, the development of trust and partnerships between business and IT, the understanding of IT's value by top management, the deep reciprocal strategy formulation process that includes environmental scanning, and CIO being knowledgeable about emerging changes in the business environment and their effect on business.

Luftman et al. [14] studied the factors that inhibit the alignment between business and IT. According to Table 1, and considering Inhibitors 1, 4, 5, 6, and 11 as intangibles, approximately 47% of business executives and 52% of IT executives consider not having good levels of intangible assets as a great inhibitor of alignment. Indeed, the lack of a relationship between business and IT executives is considered the main factor that does not allow alignment (17.6% for business executives and 21.1% for IT executives).

On the other hand, approximately 9% of IT executives and 10% of business executives consider IT a poor strategic approach (Inhibitors 7, 10, and 13) as an alignment inhibitor.

Alaceva and Rusu [15] determined 19 barriers to business-IT alignment. Among the barriers that correspond to the social dimension are a low level of understanding of the counterpart environment, resistance to sharing knowledge, the business taking decisions separately, outsourcing strategies, lack of frequent and direct formal meetings, feelings of mistrust and lack of openness, business executives and IT not being involved in each other's strategic planning, and IT management lacking leadership. They made recommendations on how to improve business-IT alignment using social-barrier mitigators, such as the use of social networks within an organization for collaborative learning.

**Table 1.** Alignment inhibitors [1].

| Inhibitors | Business Executives [2] | IT Executives [2] |
|---|---|---|
| 1. Relationship not close to IT | 17.60 | 21.10 |
| 2. IT badly prioritizes workload | 17.20 | 16.50 |
| 3. Commitments not fulfilled by IT | 15.00 | 13.30 |
| 4. IT does not understand business | 10.40 | 11.10 |
| 5. No executive support for IT | 9.10 | 11.10 |
| 6. IT management lacks leadership | 8.00 | 8.00 |
| 7. Strategic objectives not fulfilled by IT | 7.20 | 6.10 |
| 8. Budget problems | 3.50 | 2.40 |
| 9. Inadequate infrastructure | 2.20 | 1.70 |
| 10. Undefined objectives and vision | 2.00 | 1.10 |
| 11. IT does not communicate well | 1.70 | 0.90 |
| 12. Resistance from senior executives | 1.50 | 0.90 |
| 13. Untied business and IT plans | 0.90 | 1.70 |
| 14. Others, not specified | 3.50 | 3.70 |

Note: [1] Excerpted from "Enablers and inhibitors of business—IT alignment", by Luftman, J., Papp, R., and Brier, T.—Figure 3, March 1999. Used with permission from the Association for Information Systems, Atlanta, GA; 404-413-7444; www.aisnet.org. All rights reserved. [2] Affirmative response (%) calculated from the excerpted figure.

Luftman [7] created a five-level state-growth framework that describes the evolution of IT-business strategic alignment in an organization, from the least strategically aligned state to the most strategically aligned. This framework is called the Strategic Alignment Maturity Model (SAMM).

In SAMM, each of the five levels is completely described by six categories that represent management activities or practices that are critical in enabling or inhibiting strategic alignment. These categories are the following:

1.  Communication that encourages knowledge-sharing.
2.  Competency and value measurement that demonstrates the business value of IT.
3.  Governance that ensures formal discussion, the review of priorities, and the allocation of resources by the business and IT communities. Governance is the degree to which the authority for making IT decisions is defined and shared among management, and how managers in both IT and business areas set IT priorities and allocate IT resources.
4.  Partnerships that address how business and IT communities perceive each other´s contributions and their level of trust, and the way in which risks and rewards are shared between the communities.
5.  Scope and architecture that address the extent to which IT is able to drive or enable transformation, provide strategic solutions, implement flexible IT infrastructures, and enable changes to business processes for competitive advantage.
6.  Human-resource skills that enhance organizational culture and the social environment as a component of organizational effectiveness.

The social dimension involves "relationships and cognitive linkages" between business and IT, such as relationships, communications, mutual understanding, trust, respect, cultural issues, and informal structures [16] (p. 5056). Important aspects are considering the strong relationship between CEO and CIO, the involvement of IT executives in the strategic formulation of the organization, and informal relationship networks. Interpersonal communication skills and human relationships are forces that drive the creation of business-IT alignment.

Liang, Wang, Xue, and Ge [17] indicated that both intellectual and social alignment occur at the same time and have a joint effect that determines how IT alignment affects organizational agility. They defined social alignment as the shared understanding between business and IT executives that facilitates emergent co-ordination between business and IT. Social-alignment factors, such as the characteristics of the actors and the communication used in strategy formulation, lead to a mutual

understanding of these strategies. They believe that it is necessary to build communication channels and formal knowledge systems so that business and IT executives freely exchange information and opinions.

### 2.2. Intagible Assets

The notion of intangible assets was introduced in the management field in the early 1990s by Hall [18], who classified intangible resources that produce capability differentials as "assets" or "skills". Intangible assets include: (a) intellectual-property rights like trademarks, patents, copyrights, and registered designs; (b) contracts or agreements that are legally enforceable and regulated by law; (c) trade secrets or confidential information about technical and commercial aspects that are protected by law; (d) reputation, which represents the knowledge and emotions held by individuals about products or the company as a whole, and is considered a major factor in achieving competitive advantage; and (e) networks, which are those personal relationships that transcend the requirements of an organizational structure or commercial relationships, and within which people share information and gain mutual advantages.

Intangible resources are skills, and include the expertise of employees, suppliers, distributors, and culture; culture is composed of beliefs, knowledge, attitudes, and customs to which individuals are exposed in an organization, as a result of which individuals acquire a language, values, behavioral habits, and thoughts.

Intangible assets, sometimes called intellectual capital, have had many classifications and frameworks over the years, but there has been a general convergence toward three components of intangible assets: human capital, organizational or structural capital, and relational capital [19]. Human capital relates to the skills, aptitudes, and attitudes of the organization's employees or human resources; relational capital refers to the nature of the organization's relationships with all its key stakeholders; and structural capital is divided into culture, innovation, process, intellectual property, and the organization's routines and practices [20]. Human capital acts as a stimulus to the other forms of capital [21].

### 2.3. Intagible Assets, Capabilities, Knowledge, and Business-IT Strategic Alignment

The resource-based view suggests that not only is the exploitation of existing resources and competencies a means to sustained competitive advantage, but intangible assets, skill acquisition, and organizational learning that lead to knowledge stocks are also ways to stay competitive [22].

Resources are stocks of available factors that are owned or controlled by the organization, including information, systems, technology, skills, and knowledge [23]. Competencies are the capacity of the organization to deploy resources using processes, practices, and structures to affect a desired end [23]. Capabilities are the strategic application of competencies to accomplish organizational goals [23]. Capabilities arise when resources are embedded into organizational routines to form unique, firm-specific capabilities that are not readily susceptible to competitive erosion [24]. Resources can be combined and integrated into unique clusters that enable distinctive abilities within the firm [22].

This dynamism is oriented to produce strategic outcomes using and co-ordinating identified competencies. Dynamic capability is the ability to deploy superior and new configurations of functional competencies by sensing the environment, generating new knowledge, co-ordinating activities, and integrating resources [25].

Knowledge is defined as "a justified personal belief that increases an individual´s capacity to take effective action" [26]. Knowledge management is aimed to help people to transform their way of thinking and to develop initiatives that could contribute to the exploratory and exploitative organizational learning [27].

In order to implement knowledge-based strategies and upgrade existing knowledge stocks, a firm needs the capability to manage its knowledge resources [28]. Knowledge-management capability

has four processes: (a) knowledge creation [29], (b) knowledge transfer [30,31], (c) knowledge integration [32,33], and (d) knowledge leverage [34].

Knowledge creation requires the capability to generate new applications from existing knowledge resources, as well as to exploit the unexplored potential of new technologies [35]. Knowledge transfer is the extent to which business units can exchange and use each other´s knowledge and learning [36].

Knowledge integration refers to the degree to which a business unit synergistically integrates incoming knowledge resources of other business units [28] with its existing knowledge. The firm provides conductive social settings where individuals, teams, and business units share and internalize each other´s knowledge resources [29,33,37]. Knowledge exploitation requires sharing relevant knowledge among members of the firm [34]. Knowledge leverage is when the firm can ultimately use knowledge resources effectively [38].

The point of view of dynamic capabilities is that knowledge management requires two capabilities: absorptive capability [39] and high-order learning [40]. Absorptive capability is dynamic capability pertaining to knowledge creation, and it is used to enhance a firm's ability to gain and sustain competitive advantage [41]. Constant enhancement of the current knowledge stock leads by learning [42,43].

Another important dynamic capability that leads to planning strategy and alignment is the co-ordination of complex activities, which is a significant capability that an organization can develop [44]. To facilitate co-ordination, it is critical that an organization possesses superior abilities for forging internal partnerships and relationships [45–47].

This theory of co-ordination, called combinative capability [35], is present when processes that are designed by any organization are dependent on the co-ordination mechanism that assists in managing dependencies among various business functions [48,49]. Combinative capabilities are organizational integration mechanisms that enable knowledge assimilation from the outside by the whole organization so that this knowledge can be transformed and exploited [39,50].

Firms attempt to create a knowledge-conductive approach through organizational mechanisms associated with combined capabilities [50,51].

Organizations have to create conducive environments for employees to voluntarily participate in their pursuit for knowledge sharing and organizational learning, and also to cultivate a corporate culture in establishing social structures for knowledge sharing through quality interactions and meaningful discussions [52].

In organizations with strong capabilities, the executives would be able to exert great influence due to stronger available capabilities to seek, absorb, analyze, and interpret events, trends, and information [53].

### *2.4. Social Networks*

A social network is a set of interconnected people who directly or indirectly interact and influence each other [52]. A social network is a pattern of ties linking a defined set of persons or social actors [54]. Social-network ties enable collaborative work and the sharing of ideas, information, and knowledge between members of an organization [55].

Social-network structures condition the quantity and quality of knowledge acquisition, motivating the development of knowledge-management mechanisms to evaluate and integrate knowledge in the firm [56].

## 3. Conceptual Model

Intangible assets in the process of alignment between business and IT strategies need to be explored in more detail to identify which types of intangible assets are important in this process. The behavior of intangible assets in the alignment process between business and IT strategies has not been accurately explored, nor has the relationship of the various types of intangible assets in the alignment process been identified.

An organization's intangible assets, such as organizational knowledge, learning, leadership, and social networks among executives, are not jointly considered as factors that promote strategic alignment. Aspects such as the degree of interaction between groups and organizational areas that use IT, knowledge resources, capacities of multiple actors, and organizational capabilities for integration and co-ordination are also mostly not taken into account.

After a thorough literature review about the intangible factors that lead to strategic business-IT alignment, comprising the conceptualization, characterization, categorization, and relationship and organization of these categories of factors, the following hypothesis and constructs for a conceptual model are presented.

### 3.1. Hypothesis

The size, cohesion, and centrality of the network, and the strength of its ties, are characteristics of social networks [57]; due to business- and IT-executive social networks being internal and localized in the top management level of organizations, characteristics taken into account are the strength of ties, and CIO centrality in these social networks.

A social network can play a key role in enhancing organizational capabilities [58,59]; specifically, strong ties facilitate the exchange of high-quality information and tacit knowledge [58,60,61], and affect the willingness and motivation of individuals to invest time, energy, and effort in sharing knowledge with others [62], supporting the combinative capabilities of knowledge transfer and absorptive capability.

Social networks with strong ties, characterized by trust and co-operation, produce complex and tacit knowledge [56], supporting knowledge-creation capabilities. The capability of IT personnel to have strong ties with the rest of the business is considered essential to achieve a stronger fit between business and IT strategies [63]. Therefore, the first hypothesis can be defined as follows:

**H1:** *Business- and IT-executive social-network tie strength positively affects combinative capabilities to enable business-IT strategic alignment.*

On a microlevel, network measures are centrality and cohesion [62,64]; these are measurements of position within the network, and they reflect an individual's degree of prestige [56]. In this study, the characteristic of CIO centrality and leadership for partnership with business is considered. One of the factors that Brown and Magill [65] found for business-IT strategic alignment was the CIO's leadership role and partnership with business. De Haes [5] presented a list of IT government practices that influence business-IT alignment, and one of these practices was IT leadership. Luftman, Papp, and Brier [14] stated that enablers to the alignment of business and IT strategies include IT demonstrating leadership.

To facilitate internal co-ordination or combinative capability, it is critical that an organization possesses superior abilities to forge internal partnerships and relationships [45–47]. Therefore, the second hypothesis can be defined as follows:

**H2:** *CIO centrality in a business- and IT-executive social-network positively affects combinative capabilities that enable business-IT strategic alignment.*

Absorptive capability is the ability to acquire and assimilate knowledge; acquire means scanning for new knowledge, and assimilate is an understanding of what can be done with this new knowledge [66]. The major component of a firm's absorptive capability with respect to IT is represented by the conjunction of IT- and business-related knowledge, processed by, and exchanged between, IT and line managers, which includes shared knowledge and knowledge integration [67].

Combinative capabilities can help a firm disseminate and integrate new information and knowledge across the firm, and achieve the full potential of its key actual resources by using that combination [32]. The social mechanisms of integration lower barriers between knowledge assimilation and transformation, thus increasing absorptive capability [41]. These mechanisms could

encourage strategic consensus and influence the building of a shared understanding of strategic priorities [56]. Combinative capabilities are an organizational determinant of the level of a firm's absorptive capability [51].

Learning capability is the capacity within an organization to maintain or improve performance based on experience [68], as well as error detection and correction [69]. The social mechanisms of integration influence social interactions and the knowledge processes that take place between organizational members [70]. Combinative capabilities affect knowledge processes in an organization, including learning capabilities, so the third hypothesis can be defined as follows:

**H3:** *Combinative capabilities positively affect knowledge-management capabilities to enable business-IT strategic alignment.*

Kearns and Sabherwal [71] found that an organizations' emphasis on knowledge management had a positive effect on top managers' IT knowledge, which, in turn, promoted collaboration between business and IT managers who participated in the planning process of their respective counterparts. In addition, information sharing and knowledge creation facilitated business-IT strategic alignment.

Dynamic IT strategy planning is the ability to incorporate IT with business planning, and there is a significant relationship between IT knowledge management and IT technology management with dynamic IT strategy planning [35]. Knowledge that is inextricably linked with the learning process is considered a significant resource antecedent to capabilities or alignment [33,72–74]. Therefore, the fourth hypothesis can be defined as follows:

**H4:** *Knowledge-management capabilities positively affect the level of business-IT strategic alignment.*

As shown in Figure 1, there is a direct relationship between knowledge-management capabilities and the level of strategic alignment between IT and the business. At the same time, there is an indirect relationship between social-networking features of IT and business executives (strength of ties and IT manager centrality) with the level of strategic IT-business alignment, through the effect of these social networking features in the combinative capabilities of the business and IT units, and the corresponding effects of these combinative capabilities on knowledge management capabilities.

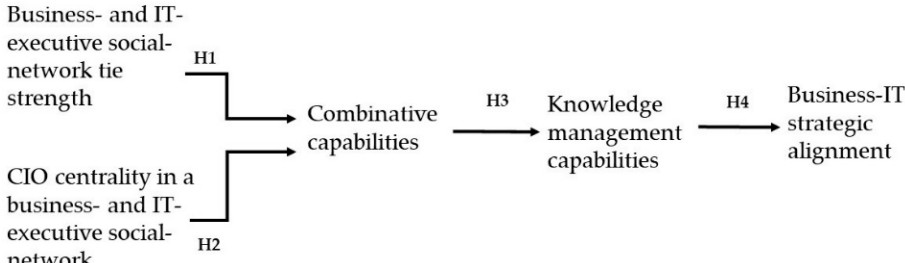

**Figure 1.** Conceptual research framework.

In Figure 1, the hypotheses and variables are as follows: (a) strength of ties in the social network of IT and business executives; (b) centrality of the IT manager in the mentioned social network; (c) combinative capabilities; (d) knowledge-management capabilities; and (e) strategic alignment between business and IT as a dependent variable.

### 3.2. Structural Model

The operationalization of the model variables based on the research carried out is presented in Table 2.

**Table 2.** Operationalization of the Structural Model Variables.

| Construct | Operational Variables | | Theoretical Support |
|---|---|---|---|
| CIO centrality in the business and IT executives' social network [62,64,75] (CENT). | a.<br>b.<br>c. | CIO's level of collaboration with other executives (CPOS).<br>Number of times the CIO is chosen by business managers in their social network (CNOM).<br>CIO's level of leadership in partnership with business (LPAR). | The central position of an individual implies a role of collaboration with other colleagues due to the exchange of information that the individual provides from their occupied place in the social structure [75]. The number of nominations that one node receives from other nodes is a measure of centrality [75–77]; the number of times an individual is chosen by colleagues in a specific social network is also a measure of centrality [56]. CIO leadership represents a position within the social network between IT and business managers, and a degree of prestige. It is also a factor for business-IT strategic alignment when the CIO has a leadership role in partnership with business [65]. |
| Strength of ties in business-IT social network [4,14,78,79] (STRT). | a.<br>b.<br>c.<br>d. | Communication between business and IT executives (COMM).<br>Business-IT executive commitment (COMI).<br>Level of trust between business and IT executives (TRUS).<br>Business-IT executive partnership (PART). | Strength of ties is operationalized as the frequency of communication and intensity of trust in a relationship [62,80–82].<br>Communication and collaboration are rooted in a sense of trust amongst team members (business and IT executives) [83] |
| Combinative capabilities between business and IT managers [56,67,84–87] (COMB). | | | |
| a. Social integration of business and IT management teams (SINT). | i.<br>ii.<br>iii.<br>iv. | Senior-level executive support to IT.<br>Business and IT managers get along well.<br>Business and IT managers are always prepared to work together and support each other.<br>There is a great deal of co-operation between business and IT managers. | Fernandez, Garcia, and Bustinza [56] measured combinative capabilities using four constructs: top manager-team social integration, co-ordination capabilities, system capabilities, and socialization capabilities. The final measure of each observed variable is anticipated to be the average of items that compose the variable [51,88–93]. |
| b. Co-ordination capabilities (COOR). | i.<br>ii.<br>iii.<br>iv. | Business and IT managers co-ordinate information sharing in a knowledge network for alignment purposes.<br>There are cross-functional teams between business and IT executives to exchange knowledge.<br>IT and business concurrently develop the same integrated planning process.<br>There is regular job rotation between business and IT executives. | Liang, Wang, Xue, and Ge [17] mentioned that co-ordination orchestrates the sequence and time in which interdependent actions should be given, the management of simultaneous activities, the exchange of information, and the mutual adjustment of actions; co-ordination can be formal structural agreements or emerging informal processes. They also indicated that previous research considered that IT alignment requires co-ordination efforts, but that they have not studied them as an explicit construct and their relationship with IT alignment. |
| c. Structural capabilities (structural capital) (STRU). | i.<br>ii.<br>iii.<br>iv. | There are formal methodologies, procedures, and tools for strategic alignment (business-IT strategic planning, enterprise architecture).<br>There are written records of performance indicators in business-IT strategic alignment.<br>Good prioritization of IT projects (business priorities are considered).<br>The CIO reports to the CEO. | In this study, these four constructs are also used to identify and examine the effect of social networks between IT and business managers for business-IT strategic-alignment purposes; the resources and knowledge captured by these managers in their social networks are distributed and integrated between managers and their organizational units. These constructs capture the social integration of business and IT management teams, co-ordination capabilities, structural capabilities (derived from structural capital), and socialization capabilities. |

**Table 2.** *Cont.*

| Construct | Operational Variables | Theoretical Support |
|---|---|---|
| d. Socialization capabilities (SCAP). | i. There is ample opportunity for informal talks between business and IT managers.<br>ii. Business and IT managers feel comfortable calling each other when needed.<br>iii. Business and IT managers are accessible to each other. | |
| Knowledge-management capabilities can be measured using outcomes as the dominant criterion [94] (KMAG). | a. Business and IT managers create knowledge about strategic alignment (KCRE).<br>b. Business and IT managers transfer knowledge about strategic alignment between each other (KTRA).<br>c. Business and IT managers integrate (share) knowledge about strategic alignment between each other (KSHA).<br>d. Business and IT managers use the shared knowledge about strategic alignment (satisfaction and enhancement of understanding) (KLEV).<br>e. Learning capability (LEAR) between business and IT managers [28,40]. LEAR is operationalized as follows:<br><br>i. There are communication channels (discussion groups, forums) to learn from previous alignment activities.<br>ii. Past mistakes in alignment activities are well-documented and understood.<br>iii. Previous bad experiences help prevent mistakes in new alignment decisions.<br>iv. Similar mistakes in alignment activities are seldom repeated.<br>v. Lessons learned from past alignment activities are used to change policies and processes. | Outcomes come from the processes involved in knowledge management: knowledge creation, knowledge transfer, knowledge integration, and knowledge leverage [29–34]; in the case of knowledge leverage, the observed variables applied to absorptive capability are the degree of satisfaction with the incoming knowledge, and the degree of enhancement of the current understanding with incoming knowledge [28]. Absorptive capability between business and IT managers [28,40,56].<br>The knowledge-management process has two capabilities. First, absorptive capability, or the firm's ability to identify, assimilate, and exploit knowledge from the environment [95]. Absorptive capability also refers to the capability to acquire and assimilate knowledge [66], where "acquire" is related to scanning new and potential knowledge, and "assimilate" is understanding what can be done with new and potential knowledge [28].<br>The second capability is learning, which is the ability to successfully apply lessons learned from previous experiences towards future initiatives [28]. Teng, Jain, and Nerur (as cited in Jain [40]) determined the precedent measures for learning capability, and Tanriverdi [28] presented an additional measure about the degree of change in policies and processes based on previously learned lessons.<br>Organizational barriers have negative influence on knowledge management adoption, specifically one of them, the lack of top management commitment. The top management must give a clear vision and create an atmosphere where knowledge sharing is encouraged [96]. Several enablers of knowledge management are leadership, quick knowledge sharing, elimination of distrust, open communication channels, and willingness to share information [97]. Top management support must encourage knowledge sharing in order to use knowledge effectively [98]. Knowledge sharing process has barriers—two of them are self-interest and divergent aspirations. Knowledge is a source of power and executives think that if it is shared to others, their importance or privileges may diminish (self-interest) [99]. Also, executives may have their own interests, incompatible with knowledge processes they are supposed to implement (divergent aspirations) [99,100]. |

**Table 2.** *Cont.*

| Construct | Operational Variables | | Theoretical Support |
|---|---|---|---|
| Business-IT strategic alignment is quantified using scale factors Strategic Orientation to Business Strategy (STROBE) [101], and Strategic Orientation to Information Systems (STROIS) [102] (SALIG) | a. | Information System (IS) support for aggressiveness through marketplace actions (AGGR). | STROBE [101] proposed eight characteristics to measure business strategy, and the STROIS [102] proposed parallel items to STROBE characteristics. STROBE and STROIS are used to determine whether used IS facilitate a business' strategic orientation. Variable business-IT strategic alignment can be operationalized using the STROBE and STROIS metrics through internal consistency between these two metrics, for instance, low grades for STROBE and high grades for STROIS indicate poor alignment [67]. STROBE and STROIS only measure implemented strategies. |
| | b. | IS support for analysis of business situations (ANAL). | |
| | c. | IS support for internal defensiveness through improving efficiencies in company operations (DEFE). | |
| | d. | IS support for external defensiveness to strengthen marketplace links (DEFE) | |
| | e. | IS support for futurity through forecasting and anticipation purposes (FUTU). | |
| | f. | IS support for proactiveness through expedited instruction of products and services (PACT). | |
| | g. | IS support for riskiness when making business risk assessments (RISK). | |
| | h. | IS support for innovation facilitating creativity and exploration (INNO). | |

The structural model with the names of the constructs defined in Table 2 is presented in Figure 2.

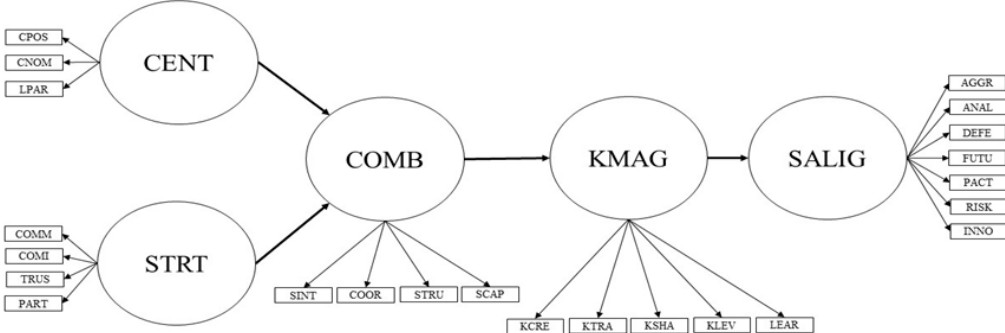

**Figure 2.** Structural model.

## 4. Discussion

Intangible assets play an important role in constructing business-IT strategic alignment. Business and IT executives' knowledge and skills, human capital, and relationships that belong to business and IT executives are considered determinant, in addition to the structural capital of the organization, which concerns processes and routines for entrenched strategic planning between business and IT.

Sharing information and creating new knowledge facilitates business-IT strategic alignment; knowledge is needed in some combination with dynamic capabilities and different resources in order to generate factors to achieve strategic alignment. The shared knowledge of business and IT executives, and capabilities that encourage and compose knowledge management, are the main factors used to explain business-IT strategic alignment. Knowledge management and co-ordination are two main capabilities in order to improve business-IT strategic alignment.

From the point of view of intangible assets, relational capital is represented by business and IT executives' social-network characteristics, structural capital by capabilities, including knowledge management and combinative capabilities, and human capital by business and IT executives' knowledge.

This conceptual model considers two characteristics of business and IT executives' social networks. First, the strength of ties that can be observed by communication and trustworthiness links, and second, the centrality that represents the CIO´s main participation and leadership in this social network.

Social-network characteristics directly affect combinative capabilities, which refer to those that promote business-IT strategic alignment, such as top management support, co-ordination between business and IT managers, structural support for business-IT alignment (for example, methodologies, tools, good prioritization of IT projects, ample opportunity for informal talks), and socialization between business and IT executives (informal connections).

Combinative capabilities affect knowledge-management capabilities, which are represented by absorptive and learning capabilities. Absorptive capability is mainly observed by knowledge shared between business and IT executives, and the use of business and IT executives' knowledge orientation to strategic alignment. Learning capability recognizes and uses past negative results as feedback in order to correct actions and fix deviations from goals with respect to business-IT strategic alignment.

## 5. Conclusions

Based on the reviewed literature, it was possible to identify the influence of several factors on social relationships between business and IT executives as related to knowledge and dynamic capabilities in an organization that may affect the level of strategic business-IT alignment. These factors are grouped within the organization, such as specific and purposeful human relationships, existing information and knowledge, and the design of an organizational structure and dynamic capabilities in order to obtain strategic business-IT alignment as superior capability.

In this model, the new proposition takes intangible assets that are related to strategic business-IT alignment, and configures them in a social network and dynamic-capabilities schema as a combination of factors in order to explain this alignment, considering a joint social-network approach with knowledge-management capabilities.

Based on existing theories and research objectives, the model identifies the independent variables that affect the dependent variables; these dependent variables can become independent in a subsequent relationship, creating the interdependent nature of the structural model. The proposed construct´s relationships will be translated into a series of structural equations for each dependent variable. This model also includes multiple constructs, each one represented by several measured variables, and these constructs are distinguished by whether they are exogenous or endogenous.

*Future Work*

Qualitative analysis of the proposed model will be conducted in order to clarify, assure, or modify elements of the conceptual model. This analysis will include interviews with business and IT executives, and the observation of events and activities from representative participants. The personal interviews will contain open-ended questions that will be recorded with permission. The observation of events will be described in an orderly and chronologically written format.

After qualitative analysis, the resultant structural model can be validated using a comprehensive multivariate technique that simultaneously examines a set of dependent relationships. This technique could be used to build a quantitative model with multiple equations.

First, the population will be enterprises from diverse sectors in Peru, the unit of analysis will be managers, deputy managers, and heads of business and IT of these enterprises, with two types of respondents: IT and business managers. These institutions perform in the same business and legal context and environment, so this situation limits the variance of the study to the internal differences identified in the data, and in the design and performance of their organizational capabilities.

The sample will be obtained by considering all public and private registered enterprises in Peru with more than 100 employees. The aim is to include a group of 150 to 200 enterprises, including three to five managers per firm.

Second, a survey is the most appropriate technique to collect data to proceed with the study. These surveys will contain a group of questions that focus on the observed independent, moderating, and dependent variables that were operationalized in the conceptual model.

Two types of questionnaires will constitute the surveys, one for each type of respondent (business or IT managers). The questions will be the same but directed toward the type of position the participant holds; for instance, business managers will be questioned on their perceptions of their relationships with IT managers, and vice versa.

The questionnaires and surveys will be piloted in a group of enterprises to be studied in order to validate the instruments in the Peruvian reality, and even more if scales should be modified with regard to some characteristics (number of points, semantic scales, or use of metric scales) for context. The validity and reliability coefficients of the final instruments will be presented.

Considering the observed variables defined for each construct, examples of questions that could be used are the following:

- "On average, how often do you communicate with business or IT managers?" The phrasing is dependent on who is being surveyed, a business or IT manager. The response is elicited on a five-point scale, where 1 means "very infrequently" and 5 means "very often".
- "To what extent do you actually engage in or support knowledge integration (sharing) about strategic business-IT alignment with business managers or IT managers?" The response is elicited on a five-point scale, where 1 means "to a very small extent" and 5 means "to a very large extent".
- "To what extent do you agree or disagree with the support for innovation facilitating creativity and exploration as it relates to information systems that are currently in production?" The response is elicited on a five-point scale, where 1 means "strongly disagree" and 5 means "strongly agree."

Third, as mentioned before, a multivariate technique for data analysis is considered due to the theoretical base developed, the existence of multiple relationships between dependent constructs, and the existence of exogenous and endogenous variables in the conceptual model. Model constructs can also be represented by observable variables, and they are indirectly measured by examining consistencies among multiple observed variables that are gathered through collected survey data.

Depending on the collected survey data, either the Structured Equation Modeling (SEM) or Partial Least Squared (PLS) statistical multivariate techniques could be used. Briefly, the common conditions for SEM analysis are the following [103]:

1. Reflective measurement—the construct explains observed variables.
2. No cross-loadings—an observed variable is only explained by its associated construct.
3. No covariance—there is no covariance between error terms associated with observed variables within a construct and between constructs.
4. Recursive model—there are no feedback loops between its constructs.
5. First-order latent factor—the model only has one layer of latent constructs.
6. Solution of a missing-data problem, specifically if the missing data follow a nonrandom pattern or represent more than 10% of the data.
7. Considering this conceptual model, a sample with more than 150 valid cases is required due to the model having seven or fewer overidentified constructs. A construct is overidentified when it has more than three observed variables [103].

If some requirements are not met (data normality, sample size, reflective measurement, first-order factor, or recursive model), it would be necessary to use the PLS estimation approach.

**Author Contributions:** M.T.-M. performed the research, literature review, conceptual model, and wrote the article. A.U.J. supervised the research and article as part of a doctoral program.

**Funding:** This research received no external funding.

**Conflicts of Interest:** The authors declare no conflict of interest.

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
