# Peer review of "An Intangible-Asset Approach to Strategic Business-IT Alignment"

_systems, doi:10.3390/systems7010017_

Reviewer 1 Report

This manuscript is about a planned study of the relationship between business management and IT management in Peruvian financial institutions, commercial, services and industrial firms. It is planned to use a questionnaire covering business and IT managers, and to analyse the responses using multivariate statistical analysis to test four hypotheses about how well business and IT strategies are aligned. No examples of the questions to be asked in the questionnaires are given, or explanation of how the analysis is proposed to be done using “Structured Equation Modeling”. If this planned study is ever done perhaps the results may be publishable, but at this stage I consider the manuscript is not publishable.

The manuscript is extremely poorly written. It does not even define what is meant by ”intangible assets”, or “strategic alignment”, terms which appear in the title. In lines 117-118 some examples of “intangible assets” are mentioned, but they are so vague and amorphous that I question how they can be useful.  Many vague or meaningless terms are used in the text without any explanation, including “combinative capabilities” (lines 16, 131, 138, 147, 151, 157, 162, 167, 169, 186, 187, 191, 198, 269, 273), “social network” (lines 14, 18, 76, 94, 118, 124, 125, 127, 128, 133, 137, 150, 184, 186, 190, 191, 198, 263, 265, 268), “knowledge management” (lines 20, 169, 171, 176, 180, 182, 187, 191, 198, 262, 263, 273),  “dynamic ” (lines 11, 175, 177). A “structural model” is depicted in Figure 2, where many entities (most undefined in the text) and relationships are shown (few explained in the text), similar to the influence diagram of System Dynamics modeling, but as the variables are mostly unquantifiable I find it hard to understand how this model can be in any way useful.

Section 2 is a “Literature Review” while Section 3 is “Background of the Problem”.  The latter discusses references [9] and [6], why does this not belong in Section 2?

In lines 63 and 64 two “current practices” are mentioned, (a) communication between business and IT executives, and (b) connection between business and IT planning. Are these really different? The manuscript is full of “motherhood statements”, eg lines 108-113, 128-132, 226-229.

In summary, this manuscript is poorly written, full of “motherhood statements” and meaningless jargon and is far too premature for publication at this stage.

Author Response

Response to Reviewer 1 Comments

Comments and Suggestions for Authors

Point 1: This manuscript is about a planned study of the relationship between business management and IT management in Peruvian financial institutions, commercial, services and industrial firms. It is planned to use a questionnaire covering business and IT managers, and to analyse the responses using multivariate statistical analysis to test four hypotheses about how well business and IT strategies are aligned. No examples of the questions to be asked in the questionnaires are given, or explanation of how the analysis is proposed to be done using “Structured Equation Modeling”. If this planned study is ever done perhaps the results may be publishable, but at this stage I consider the manuscript is not publishable.

Response 1:

The article was reformulated in order to present the conceptual model developed as a result of the article. The section “Future work” has been adjusted to outline the further activities in order to make a second phase of the research, build a quantitative model based on the conceptual model presented in the article.  

In section “Future work” some examples of scale of the measures for the model´s observed variables have been presented. Each observed variable defined for each construct of the model is measured with a five or seven-point scale, that represents the perception of the business or IT executive about this observed variable. Examples of questions that quantify some observed variables are the following:

·         “On average, how often do you communicate with business/IT managers?” The phrasing is dependent on who is being surveyed, a business manager or an IT manager. The response is elicited on a seven-point Likert-type scale for communication frequency, where 1 means “very infrequently” and 7 means “very often”.

·         “To what extent do you actually engage in or support the knowledge integration (sharing) about business-IT strategic alignment with business managers/IT managers?”. The response is elicited on a five-point scale, where 1 means “a very small extent” and 7 means “a very large extent”.

·         “To what extent do you agree or disagree with the support for innovation facilitating creativity and exploration as it relates to the information systems that are currently in production?”. The response is elicited on a five-point scale, where 1 means “strongly disagree”, 2 “disagree”, 3 “neutral”, 4 “agree”, and 5 “strongly agree.”

In section “Future Work” is proposed a mixed research method to develop the structural model, interviews with IT and business executives will be conducted to clarify, assure, or modify some element of the conceptual model (previous small-scale study). These executives will be purposefully sampled. Also, observation of events and activities for some participants will be included.

After the qualitative analysis, the resultant structural model can be validated using a statistical multivariate technique that fulfil the statistical characteristics of the model and the data collected. The quantitative analysis will be made with surveys and questionnaires that quantify the level of observed variables as a perception of the executives surveyed, using a seven or five scale points.

Depending on the conditions that data collected in the surveys, either the Structured Equation Modelling (SEM) or Partial Least Squared (PLS) statistical multivariate techniques could be used.

If some requirements of SEM are not met (data normality, sample size, reflective measurement, first order factor, or recursive model), it will be necessary to use the partial least square (PLS) estimation approach.

Point 2: The manuscript is extremely poorly written. It does not even define what is meant by ”intangible assets”, or “strategic alignment”, terms which appear in the title. In lines 117-118 some examples of “intangible assets” are mentioned, but they are so vague and amorphous that I question how they can be useful.  Many vague or meaningless terms are used in the text without any explanation, including “combinative capabilities” (lines 16, 131, 138, 147, 151, 157, 162, 167, 169, 186, 187, 191, 198, 269, 273), “social network” (lines 14, 18, 76, 94, 118, 124, 125, 127, 128, 133, 137, 150, 184, 186, 190, 191, 198, 263, 265, 268), “knowledge management” (lines 20, 169, 171, 176, 180, 182, 187, 191, 198, 262, 263, 273),  “dynamic ” (lines 11, 175, 177). A “structural model” is depicted in Figure 2, where many entities (most undefined in the text) and relationships are shown (few explained in the text), similar to the influence diagram of System Dynamics modeling, but as the variables are mostly unquantifiable I find it hard to understand how this model can be in any way useful.

Response 2:

Intangible assets have been defined in lines 184-195. Strategic alignment has been defined in lines 72-73. Combinative capabilities have been defined in lines 245-247. Social network has been defined in lines 253-255. Knowledge management has been defined in lines 107-108 and 223-225. Dynamic capabilities have been defined in lines 219-221.

The entities of the structural model depicted in Figure 2 have been defined in Table 2. In reference to an explanation for the relationships within this structural model, a section “Discussion” has been included with the explanation of the main relationships between the constructs of the model.

The structural model has constructs (variables) that are quantifiable through observed variables, these variables get a value through the business and IT executive´s response to certain questions that are prepared as part of the future work (see Response 1, section “Future work”).

Depending on the conditions of the data collected in the surveys, either the Structured Equation Modelling (SEM) or Partial Least Squared (PLS) statistical multivariate techniques could be used, to calculate the parameters of the model and to test the statistical significance of the relationships between constructs. 

Point 3: Section 2 is a “Literature Review” while Section 3 is “Background of the Problem”.  The latter discusses references [9] and [6], why does this not belong in Section 2?

Response 3:

Previous sections 2 and 3 has been merged to get a greater and complete section “Literature Review”, with a detailed and coherent chain of concepts and structure of ideas that formulate the problem, and the model that conceptualizes the factors that encourage strategic alignment.

Point 4: In lines 63 and 64 two “current practices” are mentioned, (a) communication between business and IT executives, and (b) connection between business and IT planning. Are these really different? The manuscript is full of “motherhood statements”, eg lines 108-113, 128-132, 226-229.

Response 4:

In Section “Literature Review” lines 82 to 89 have been defined the concepts “communications between business and IT executives” and “connection between business and IT planning”.

Communication between business and IT executives is the contacts between individuals during the daily course of events: Face to face or written, formal or informal, in groups or in pairs. This concept represents all communications between IT and business executives except formal long-term planning sessions.

The connection between business and IT planning is the degree to which the business and IT planning processes are interrelated.

In summary, this manuscript is poorly written, full of “motherhood statements” and meaningless jargon and is far too premature for publication at this stage.

Reviewer 2 Report

The paper is interesting and deals with topic which is intensively studied by both academicians and practitioners. The main point (or message) is clear and the research goal is explained understandably. The context is sufficiently described.

Minor issues:

Flow of ideas is mostly clear. Occasionally, authors should reformulate specific segments of text, change word order or make other stylistic changes. Example n.1, in abstract, authors mention a model in two ways which are quite confusing. At the beginning “A conceptual model…will be defined”. Then, “A structural model is proposed”. What is the difference? Is the “conceptual model” the model presented in the paper? Then starting abstract with the achieved result is not an appropriate beginning of this piece of text. I suggest to reformulate abstract. Example n.2, some sentences are too long. Substitution of commas by full points would create two meaningful sentences which are easier to read and understand. For instance, first sentence in section 3 “…the alignment between the business and IT[. A]ccording to Table 1…”, or first sentence in section 5.1 “…and IT of these enterprises[. I]n this study, the plan…”.

While H1 contains “network”, H2 comprises “networks” – is this ok (then explain), or just typing error?

Major issue:

There is only one major issue associated with this paper. Hope my explanation is be understandable…

During reading the paper, I liked it. But then, I was quite surprised that there was the end without any results related to the outlined research. I know that authors state that their paper is theoretical. I understand it although I am not able to foresee how it can be turned into practical benefits right now. Again, that is fine, paper is theoretical and I like the conceptual model that explicitly lists (I really like Table 2) related concepts. BUT paper is presented in a form that makes it incomplete. Research results presented in journals should be complete, described from A to Z. Further research can only outline what can be done (like in section 6.1 here). I consider description of further research in a way formulated in this paper as absolutely improper. This is suitable for conference proceedings where authors present “work in progress”. Very likely unintentionally, authors confirm this at the beginning of section 6.1 “…must be completed in order to fulfil this research…”. Reformulated, this sentence states that “we did something, but it is not enough; therefore we defined a set of activities, which we have not conducted yet…”.

My opinion is that if authors want (or are willing) to change the paper to a form acceptable for journals, they need to remove section 5 (there is no sense talking about methodology of activities which are eventually not conducted) and shift paper’s nature to “conceptual paper with literature review”. I really do like support by references which is applied in text. Number of information resource is sufficient. But many are used for description of current state and are not actively used in problem formulation. Merging sections 2 and 3 into a coherent section that step by step leads to formulation of existing problem in literature would bring better flow of ideas and straightforwardness to text. This change would give a space for elaboration of existing concepts or inclusion of other (like Communities in Practice or Communities of Interest in the knowledge management context) – which is not necessary, I only propose it.

Moreover, a short section in which achieved results (model) is discussed would be beneficial for readers.

I am convinced that similar modification (not exactly this one, but in a similar nature) can change appropriateness of this paper for publication in the journal. It would offer one main added value (model) and applied methodology (literature review and conceptual analysis). With discussion of results and outline of further research, the paper would be complete and meaningful.

I am aware that reviewers should not suggest how research should have been done or what type of research would be better. To be clear, I do not want to do this. I only suggest to change the paper style with presentation of the same research results. I suggest MAJOR REVISIONS only to give authors a place for a change if they are willing to do it. I they do not like this suggestion and not reformulate the paper, I will find the revised version unacceptable (basically as this one) and will suggest REJECT as I do not consider this paper suitable for publication in its current form and style.  

The “out of review” remark: reconsider the method for further research (the part only outlined but not solved here). Based on the description, it seems to me that quantitative research supported by survey will not give you exactly what you need. It seems to me that internal validity of this research would be quite low. I am convinced that mixed research methodology would be more appropriate and bring more valuable results (i.e. include qualitative methods such as interviews or document analysis – these are sources of data and information in which you can find the truth, surveys are unreliable for similar research purposes; it gives you a chance for interesting statistical processing but it cannot reveal as many facts that qualitative research)…

Author Response

Response to Reviewer 2 Comments

Comments and Suggestions for Authors

The paper is interesting and deals with topic which is intensively studied by both academicians and practitioners. The main point (or message) is clear and the research goal is explained understandably. The context is sufficiently described.

 Minor issues:

Point 1: Flow of ideas is mostly clear. Occasionally, authors should reformulate specific segments of text, change word order or make other stylistic changes. Example n.1, in abstract, authors mention a model in two ways which are quite confusing. At the beginning “A conceptual model…will be defined”. Then, “A structural model is proposed”. What is the difference? Is the “conceptual model” the model presented in the paper? Then starting abstract with the achieved result is not an appropriate beginning of this piece of text. I suggest to reformulate abstract. Example n.2, some sentences are too long. Substitution of commas by full points would create two meaningful sentences which are easier to read and understand. For instance, first sentence in section 3 “…the alignment between the business and IT[. A]ccording to Table 1…”, or first sentence in section 5.1 “…and IT of these enterprises[. I]n this study, the plan…”.

While H1 contains “network”, H2 comprises “networks” – is this ok (then explain), or just typing error?

Response 1:

The Abstract was reformulated, only “a conceptual model….” is mentioned, and the achieved results are not presented at the beginning of this text.

All the sentences of the article have been reviewed for verifying their longitude and their understanding.

The word “networks” in hypothesis H2 was a typing error.

Major issue:

There is only one major issue associated with this paper. Hope my explanation is be understandable…

Point 2: During reading the paper, I liked it. But then, I was quite surprised that there was the end without any results related to the outlined research. I know that authors state that their paper is theoretical. I understand it although I am not able to foresee how it can be turned into practical benefits right now. Again, that is fine, paper is theoretical and I like the conceptual model that explicitly lists (I really like Table 2) related concepts. BUT paper is presented in a form that makes it incomplete. Research results presented in journals should be complete, described from A to Z. Further research can only outline what can be done (like in section 6.1 here). I consider description of further research in a way formulated in this paper as absolutely improper. This is suitable for conference proceedings where authors present “work in progress”. Very likely unintentionally, authors confirm this at the beginning of section 6.1 “…must be completed in order to fulfil this research…”. Reformulated, this sentence states that “we did something, but it is not enough; therefore we defined a set of activities, which we have not conducted yet…”.

Response 2:

Following this recommendation, the article was reformulated in order to present the conceptual model developed as a result of the article, and the practical benefits for enterprises and organizations to know about the importance and effect of the intangible assets on business-IT strategic alignment.

The section “Literature Review” has been extended to conform a section that explain from the beginning to end, the ideas and concepts that back the constructs and relationships within the conceptual model.

The section “Future work” has been adjusted to outline the further activities in order to make a second phase of the research, build a quantitative model based on the conceptual model presented in the article.  

Point 3: My opinion is that if authors want (or are willing) to change the paper to a form acceptable for journals, they need to remove section 5 (there is no sense talking about methodology of activities which are eventually not conducted) and shift paper’s nature to “conceptual paper with literature review”. I really do like support by references which is applied in text. Number of information resource is sufficient. But many are used for description of current state and are not actively used in problem formulation. Merging sections 2 and 3 into a coherent section that step by step leads to formulation of existing problem in literature would bring better flow of ideas and straightforwardness to text. This change would give a space for elaboration of existing concepts or inclusion of other (like Communities in Practice or Communities of Interest in the knowledge management context) – which is not necessary, I only propose it.

Response 3:

Following this recommendation, the section “Methodology” has been removed and the nature of conceptual paper is reaffirmed, the new methodology is presented in lines 261-264 where is mentioned that a thorough literature review guides the chain of concepts and abundant previous researches to the final version of the conceptual model; fulfilling the aim of this article with a new approach about business and IT executive´s interrelationships, organizational capabilities, and strategic alignment.

The methodology comprises the conceptualization, analysis, characterization, categorization, the organization and sorting of these categories of concepts and relationships in order to set out the complete conceptual model, and the hypothesis about the effects of intangible assets on business-IT strategic alignment.

Previous sections 2 and 3 has been merged to get a greater and complete section “Literature Review”, with a detailed and coherent chain of concepts and structure of ideas that formulate the problem, and the model that conceptualizes the factors that encourage strategic alignment.

Point 4: Moreover, a short section in which achieved results (model) is discussed would be beneficial for readers.

Response 4:

It has been included a section “Discussion” that sheds light on the results expressed as the relationships between the constructs that lead direct and indirect effects on business-IT strategic alignment.

I am convinced that similar modification (not exactly this one, but in a similar nature) can change appropriateness of this paper for publication in the journal. It would offer one main added value (model) and applied methodology (literature review and conceptual analysis). With discussion of results and outline of further research, the paper would be complete and meaningful.

I am aware that reviewers should not suggest how research should have been done or what type of research would be better. To be clear, I do not want to do this. I only suggest to change the paper style with presentation of the same research results. I suggest MAJOR REVISIONS only to give authors a place for a change if they are willing to do it. I they do not like this suggestion and not reformulate the paper, I will find the revised version unacceptable (basically as this one) and will suggest REJECT as I do not consider this paper suitable for publication in its current form and style.  

 Point 5: The “out of review” remark: reconsider the method for further research (the part only outlined but not solved here). Based on the description, it seems to me that quantitative research supported by survey will not give you exactly what you need. It seems to me that internal validity of this research would be quite low. I am convinced that mixed research methodology would be more appropriate and bring more valuable results (i.e. include qualitative methods such as interviews or document analysis – these are sources of data and information in which you can find the truth, surveys are unreliable for similar research purposes; it gives you a chance for interesting statistical processing but it cannot reveal as many facts that qualitative research)…

Response 5:

In section “Future Work” is proposed a mixed research method to develop the structural model, interviews with IT and business executives will be conducted to clarify, assure, or modify some element of the conceptual model. These executives will be purposefully sampled. Also, observation of events and activities for some participants will be included.

After the qualitative analysis, the resultant structural model can be validated using a statistical multivariate technique that fulfil the statistical characteristics of the model and the data collected. The quantitative analysis will be made with surveys and questionnaires that quantify the level of observed variables as a perception of the executives surveyed, using a seven or five scale points.

Reviewer 3 Report

Manuscript ID: systems-430005

Title: An approach of Intangible Assets on Business-IT Strategic Alignment

Authors: Miguel Tejada-Malaspina

Title of the paper: An approach of intangible assets on business-IT strategic alignment

Summary of the paper

This article presents the development of a conceptual model and a research design to analyze the impact of social factors between business and IT managers on the strategic alignment of business and IT in organizations. The factors include collaboration, trust, partnership, coordination capabilities, socialization capabilities and knowledge management capability.

The conceptual part of the paper is based on the literature on business-IT strategic alignment. Four hypotheses are formulated and a structural model is presented. A research design is presented in which the author explains their desired data collection and analysis method. Future works include the distribution and collection of electronic surveys to business and IT managers, followed by a SEM analysis.

Context and relevance of the paper

The subject of business-IT strategic alignment is still relevant in today’s organizations and the author has chosen an angle that was not fully studied in the past. Thus, the research design presented in this paper has a good potential contribution to research in the field. It could also contribute to practice trough a better understanding of the social and intangible factors influencing a good business-IT alignment.

However, the introduction section should be improved by citing relevant literature and explaining in more details the context leading to the choice of this research angle. This information is in chapter 3 but some of it could be presented earlier.

Theoretical basis, quality of the theoretical model

Excluding the introduction (as presented above) and the methodology (as discussed below) chapters, claims in the paper are supported by relevant literature.

The literature review chapter is short and does not reflect all the subjects presented in the development of the model. Notably, less common concepts such as absorptive capability and combinative capabilities should be defined earlier in the paper.

A cross-study of previous works would be an interesting addition to the literature review chapter. This would help readers to get a better view of the coverage of the different subjects in past research, given the number of theoretical concepts involved in the proposed model.

The proposed model is presented with enough details and references to understand the research hypotheses. All concepts and constructs are defined. The operationalization of the variables is presented for most constructs, although the scales of the measures are not.

Research design

The recommended research design is described, but the choices are not justified and are not backed by appropriate methodological references. Example are lines 241 to 248, where SEM is suggested but without explaining why it is more appropriate than other confirmatory methods.

The desired population and sampling method are explained with enough details considering the preliminary state of the project. The chosen population and the data collection method seem appropriate to answer the research objective mentioned at line 30. Little detail is given about the questionnaire items and their scales, outside mention that they are taken from previous studies. A pilot is mentioned in chapter 6, but it would be best to present it in chapter 5. Furthermore, a validation of the proposed model in a qualitative phase was not mentioned. Considering the proposed model is not based on a previous small-scale study (or if it is, there is no reference to it), this first step would probably contribute to a higher quality of the model.

Contribution of the paper and overall quality

The article overall is well-written. With the above-mentioned exceptions, it has good theoretical basis and addresses an issue that is relevant for research, and whose results could be relevant to practice.

A more exhaustive presentation of the theoretical concepts earlier in the paper and a better explanation early on of the relevance of the context are the main improvement points identified in this review.

Chapter 5 on the suggested methodology is the other improvement opportunity, with a better theoretical basis and the inclusion of a small scale and/ or qualitative evaluation of the proposed model the main considerations.

Author Response

Response to Reviewer 3 Comments

Comments and Suggestions for Authors

Manuscript ID: systems-430005

Title: An approach of Intangible Assets on Business-IT Strategic Alignment

Authors: Miguel Tejada-Malaspina

Title of the paper: An approach of intangible assets on business-IT strategic alignment

Summary of the paper

This article presents the development of a conceptual model and a research design to analyze the impact of social factors between business and IT managers on the strategic alignment of business and IT in organizations. The factors include collaboration, trust, partnership, coordination capabilities, socialization capabilities and knowledge management capability.

The conceptual part of the paper is based on the literature on business-IT strategic alignment. Four hypotheses are formulated and a structural model is presented. A research design is presented in which the author explains their desired data collection and analysis method. Future works include the distribution and collection of electronic surveys to business and IT managers, followed by a SEM analysis.

Context and relevance of the paper

The subject of business-IT strategic alignment is still relevant in today’s organizations and the author has chosen an angle that was not fully studied in the past. Thus, the research design presented in this paper has a good potential contribution to research in the field. It could also contribute to practice trough a better understanding of the social and intangible factors influencing a good business-IT alignment.

Point 1: However, the introduction section should be improved by citing relevant literature and explaining in more details the context leading to the choice of this research angle. This information is in chapter 3 but some of it could be presented earlier.

Theoretical basis, quality of the theoretical model

Excluding the introduction (as presented above) and the methodology (as discussed below) chapters, claims in the paper are supported by relevant literature.

Response 1:

The section “Introduction” has been improved explaining with more detail the context of the business-IT alignment studies since more than 30 years ago. Specifically, the social factors of this alignment because the purpose of this research is to determine the social factors, its relationships, and the conceptual model of these social factors that lead to business-IT strategic alignment.

Point 2: The literature review chapter is short and does not reflect all the subjects presented in the development of the model. Notably, less common concepts such as absorptive capability and combinative capabilities should be defined earlier in the paper.

A cross-study of previous works would be an interesting addition to the literature review chapter. This would help readers to get a better view of the coverage of the different subjects in past research, given the number of theoretical concepts involved in the proposed model.

Response 2:

The section “Literature Review” has been extended to conform a section that explain from the beginning to end, the ideas and concepts that back the constructs and relationships within the conceptual model.

Previous sections 2 and 3 has been merged to get a greater and complete section “Literature Review”, with a detailed and coherent chain of concepts and structure of ideas that formulate the problem, and the model that conceptualizes the factors that encourage strategic alignment.

Absorptive capability has been defined in lines 235-237 and 302-304. Combinative capabilities have been defined in lines 245-247.

Point 3: The proposed model is presented with enough details and references to understand the research hypotheses. All concepts and constructs are defined. The operationalization of the variables is presented for most constructs, although the scales of the measures are not.

Response 3:

In section “Future work” some examples of scale of the measures for the model´s observed variables have been presented. Each observed variable defined for each construct of the model is measured with a five or seven-point scale, that represents the perception of the business or IT executive about this observed variable. Examples of questions that quantify some observed variables are the following:

·         “On average, how often do you communicate with business/IT managers?” The phrasing is dependent on who is being surveyed, a business manager or an IT manager. The response is elicited on a seven-point Likert-type scale for communication frequency, where 1 means “very infrequently” and 7 means “very often”.

·         “To what extent do you actually engage in or support the knowledge integration (sharing) about business-IT strategic alignment with business managers/IT managers?”. The response is elicited on a five-point scale, where 1 means “a very small extent” and 7 means “a very large extent”.

·         “To what extent do you agree or disagree with the support for innovation facilitating creativity and exploration as it relates to the information systems that are currently in production?”. The response is elicited on a five-point scale, where 1 means “strongly disagree”, 2 “disagree”, 3 “neutral”, 4 “agree”, and 5 “strongly agree.”

 Research design

Point 4: The recommended research design is described, but the choices are not justified and are not backed by appropriate methodological references. Example are lines 241 to 248, where SEM is suggested but without explaining why it is more appropriate than other confirmatory methods.

The desired population and sampling method are explained with enough details considering the preliminary state of the project. The chosen population and the data collection method seem appropriate to answer the research objective mentioned at line 30. Little detail is given about the questionnaire items and their scales, outside mention that they are taken from previous studies. A pilot is mentioned in chapter 6, but it would be best to present it in chapter 5. Furthermore, a validation of the proposed model in a qualitative phase was not mentioned. Considering the proposed model is not based on a previous small-scale study (or if it is, there is no reference to it), this first step would probably contribute to a higher quality of the model.

Response 4:

The section “Methodology” (research design) has been removed and the nature of conceptual paper is reaffirmed, the new methodology is presented in lines 261-264 where is mentioned that a thorough literature review guides the chain of concepts and abundant previous researches to the final version of the conceptual model.

The methodology comprises the conceptualization, analysis, characterization, categorization, the organization and sorting of these categories of concepts and relationships in order to set out the complete conceptual model, and the hypothesis about the effects of intangible assets on business-IT strategic alignment.

In section “Future Work” is proposed a mixed research method to develop the structural model, interviews with IT and business executives will be conducted to clarify, assure, or modify some element of the conceptual model (previous small-scale study). These executives will be purposefully sampled. Also, observation of events and activities for some participants will be included.

After the qualitative analysis, the resultant structural model can be validated using a statistical multivariate technique that fulfil the statistical characteristics of the model and the data collected. The quantitative analysis will be made with surveys and questionnaires that quantify the level of observed variables as a perception of the executives surveyed, using a seven or five scale points.

Depending on the conditions of the data collected in the surveys, either the Structured Equation Modeling (SEM) or Partial Least Squared (PLS) statistical multivariate techniques could be used.

If some requirements of SEM are not met (data normality, sample size, reflective measurement, first order factor, or recursive model), it will be necessary to use the partial least square (PLS) estimation approach.

In Response 3 is mentioned some detail about the questionnaire items.

In section “Future work” is mentioned that questionnaires and surveys to be used will be piloted in a group of enterprises to be studied in order to validate the instruments in the Peruvian reality.

Contribution of the paper and overall quality

The article overall is well-written. With the above-mentioned exceptions, it has good theoretical basis and addresses an issue that is relevant for research, and whose results could be relevant to practice.

Point 5: A more exhaustive presentation of the theoretical concepts earlier in the paper and a better explanation early on of the relevance of the context are the main improvement points identified in this review.

Response 5:

Same as Responses 1 and 2.

Point 6: Chapter 5 on the suggested methodology is the other improvement opportunity, with a better theoretical basis and the inclusion of a small scale and/ or qualitative evaluation of the proposed model the main considerations.

Response 6:

Part of Response 4.

Round  2

Reviewer 2 Report

Thank you for implementation of my suggestion. I am convinced that restructuring of the paper increased its quality and readability. 

Author Response

Thank you, the english language and style have been checked by the editing service.

Reviewer 3 Report

In the revised version the authors made it clearer that this is a conceptual paper, and they completed their literature review with the elements requested by the reviewers. They also addressed the lack of methodological details in the “future work” part, including steps preliminary to SEM.

The added text is a less clear and of a lower language quality than the original manuscript. Revision should be considered.

Author Response

Thank you, the added text has been improved, and the english language has been checked by the editing service.